# Piezo1 ion channels inherently function as independent mechanotransducers

**Amanda H Lewis, Jörg Grandl\***

Department of Neurobiology, Duke University Medical Center, Durham, United States

**Abstract** Piezo1 is a mechanically activated ion channel involved in sensing forces in various cell types and tissues. Cryo-electron microscopy has revealed that the Piezo1 structure is bowl-shaped and capable of inducing membrane curvature via its extended footprint, which indirectly suggests that Piezo1 ion channels may bias each other's spatial distribution and interact functionally. Here, we use cell-attached patch-clamp electrophysiology and pressure-clamp stimulation to functionally examine large numbers of membrane patches from cells expressing Piezo1 endogenously at low levels and cells overexpressing Piezo1 at high levels. Our data, together with stochastic simulations of Piezo1 spatial distributions, show that both at endogenous densities (1–2 channels/$\mu m^2$), and at non-physiological densities (10–100 channels/$\mu m^2$) predicted to cause substantial footprint overlap, Piezo1 density has no effect on its pressure sensitivity or open probability in the nominal absence of membrane tension. The results suggest that Piezo channels, at densities likely to be physiologically relevant, inherently behave as independent mechanotransducers. We propose that this property is essential for cells to transduce forces homogeneously across the entire cell membrane.

## Introduction

Cells are constantly exposed to mechanical forces and have evolved diverse molecular mechanisms that enable force detection. The rapid sensing of mechanical forces that occurs in milliseconds is achieved by force-gated ion channels that convert mechanical energy into electrochemical signals (*Kefauver et al., 2020*). For example, Piezo1 is a mechanically gated cation channel that directly senses membrane tension (T) with high sensitivity (*Coste et al., 2010*; *Cox et al., 2016*; *Lewis and Grandl, 2015*; *Syeda et al., 2016*). Therefore, theoretically, any mechanical disturbance of the cell membrane may lead to changes in Piezo1 channel open probability ($P_o$). However, the propagation of membrane tension away from its source has been shown to be relatively slow (D ~ 0.024 $\mu m^2$/s), poising Piezo channels with the ability to efficiently and rapidly transduce only mechanical forces that are nearby (*Shi et al., 2018*).

The degree to which forces are transduced homogeneously across the cell membrane likely depends not only on the dynamics of membrane tension propagation, but also on (i) the spatial distribution of force-gated ion channels, and (ii) on their functional independence.

Any substantial deviation from a uniform distribution of force-gated ion channels will result in domains that fail to detect forces (where there are no ion channels) as well as domains that transduce forces disproportionately (where many ion channels are nearby). Functional interactions and/or cooperativity between force-gated ion channels may further skew transduction efficiency or kinetics in a manner that depends on local channel density, effectively creating a patchwork of blind spots and sensitized areas. Indeed, there is recent experimental evidence suggesting that Piezo1 channels may be non-uniformly distributed. TIRF imaging of Piezo1 channels revealed their free diffusion over the surface of live neural stem/progenitor cells, indirectly suggesting channels are not tethered to other static structures and that random membrane distribution may be possible. However, in the same cells, Piezo1-related activity was spatially enriched, which could arise from non-uniform distribution

**\*For correspondence:**
grandl@neuro.duke.edu

**Competing interest:** The authors declare that no competing interests exist.

**eLife digest** Cells can sense a range of mechanical forces both inside and outside the body, such as the stroke of a fingertip or the filling of a lung. Pores on the surface of the cell called Piezo channels open up in response to this pressure. This allows ions to flood in to the cell and trigger a series of biochemical reactions that alter the cell's behavior.

Piezo channels have a unique bowl-like structure that transforms the shape of the cell surface around them, potentially affecting how nearby proteins behave. Previous research had suggested that these channels might be unevenly distributed across the cell surface, and were predicted to modify each other's behaviors when tightly packed together. This cooperative response would have a significant impact on how cells sense mechanical force.

To investigate if this was the case, Lewis and Grandl studied a mouse cell called Neuro2A which naturally produces Piezo ion channels. In the experiment, pressure was applied to different parts of the cell and the electric current generated by ions moving across the surface was recorded: the higher the electrical activity, the more ion channels present. This showed that Piezo channels are randomly distributed across the cell surface and do not tend to cluster together. The same Neuro2A cells were then engineered to produce up to one hundred times more Piezo proteins. Despite the channels being more densely packed together, how they responded to mechanical force remained the same.

These results suggest that Piezo channels act independently and are not influenced by close proximity to one another. Lewis and Grandl propose that this property may ensure that all parts of the cell surface react to mechanical force in the same way. Further work is needed to see if this finding applies to other cell types that produce Piezo proteins.

of Piezo1 channels, variable membrane tension, or both (*Ellefsen et al., 2019*). STORM imaging of HEK293 cells stably expressing GFP-tagged Piezo1 also suggested a non-homogenous spatial distribution, but at channel densities that are likely not physiological (*Ridone et al., 2020*). Additional studies have proposed the existence of clusters with varying numbers of channels (*Jiang et al., 2021*), and that Piezo1 may be concentrated at focal adhesions (*Mingxi Yao et al., 2020*) and at wound edges in keratinocytes (*Holt et al., 2021*).

A compelling theoretical framework predicting spatial and functional interactions between nearby Piezo channels has been constructed based on cryo-electron microscopy structures of Piezo1, which revealed its large and extremely curved dome-shaped structure (*Guo and MacKinnon, 2017*; *Haselwandter and MacKinnon, 2018*; *Saotome et al., 2018*). Theoretical calculations based on these structures suggest that Piezo1 can both sense and curve the proximal membrane with a characteristic decay length extending ~14 nm beyond the channel. The energetic cost of this membrane curvature implies that Piezo1 channels that are within ~3 decay lengths (a 'footprint' of ~50 nm; see Materials and methods) may influence each other via multiple mechanisms: opposing curvature may cause nearby Piezo1 channels to repel each other, as well as influence each other's gating properties (*Haselwandter and MacKinnon, 2018*). Indeed, imaging of vesicles containing reconstituted Piezo1 has demonstrated the ability of Piezo1 to curve the membrane locally and suggested that flattening of the channel dome may directly couple to channel opening, but the precise effects of membrane energetics on gating have yet to be explored experimentally (*Jiang et al., 2021*; *Lin et al., 2019*).

Here, we set out to quantify the spatial distribution and potential functional interactions of Piezo1 ion channels across multiple orders of channel densities. Specifically, we use electrophysiology and large numbers of independent measurements to simultaneously quantify channel number and function, including single-channel current, open probability, and pressure sensitivity. Collectively, our electrophysiological data, together with stochastic simulations show that Piezo1 ion channels, at densities likely to be physiologically relevant, inherently function as independent mechanotransducers. We propose that this property enables cells to transduce forces with high spatial homogeneity across the entire cell membrane.

## Results

### Piezo1 function at low channel densities

To simultaneously measure and thus directly correlate Piezo1 channel density with its gating properties, we performed cell-attached patch-clamp electrophysiology on Neuro2A cells, which natively express Piezo1 (*Coste et al., 2010*). We chose Neuro2A cells over human umbilical vein endothelial cells (HUVECs; *Figure 1—figure supplement 1*) and other cell lines because they have among the highest endogenous levels of Piezo1 expression, thus offering the highest likelihood for observing any potential functional effects of local channel density on function. We then designed a novel stimulation protocol that we optimized to assess Piezo1 single-channel conductance (i), pressure sensitivity ($P_{50}$), and number of channels (n) in each patch with high accuracy. Since Piezo1 inactivation precludes a precise measurement of maximal current ($I_{max}$) at negative potentials, we recorded currents at +60 mV, where inactivation during short pressure steps is minimal (*Coste et al., 2010*; *Wu et al., 2017*). However, traditional step protocols are well known to induce irreversible changes in patch geometry owing to membrane creep, particularly at positive voltages (*Lewis and Grandl, 2015*; *Suchyna et al., 2009*). Indeed, a direct step to a saturating pressure (–60 mmHg) caused small, but measurable changes in current due to leak and/or capacitance even in patches that were later classified to have zero channels (see below; mean ± SD: 2.6 ± 1.3 pA; n=33 patches; *Figure 1—figure supplement 2A-B*). Consistent with this idea, the same pressure step caused a similarly small current in patches from Neuro2A-Piezo1ko cells, which have been CRISPR-engineered to lack Piezo1 channels (mean ± SD: 3.8 ± 1.5 pA, n=15 patches; *Figure 1—figure supplement 2C*). Consequently, a step protocol consistently overestimates current by 2–3 pA, and thus channel numbers accordingly (*Figure 1—figure supplement 2B*).

To overcome this limitation, we instead designed a protocol in which the pressure was decreased incrementally in 1 mmHg steps from 0 mmHg to –60 mmHg in 50 ms intervals, thus approximating a ramp (*Figure 1A–D*, top). The total duration of this ramp is short enough to minimize membrane creep (3 s), but each step is long enough (50 ms) to allow pressure equilibration, which occurs within ~7 ms (*Lewis et al., 2017*). Similar to the step protocol, the ramp protocol elicited small background changes in current amplitude in all patches. However, instead of the square-like current increases evoked by the step protocol, here the current changed more steadily, allowing us to perform linear background subtractions for each individual patch (*Figure 1A–D*, middle) and ultimately achieve a more reliable detection of channel gating events, which were easily discerned as nearly instantaneous changes in current amplitude (*Figure 1A–D*, top, arrows). We are confident that these slow changes in current are unrelated to Piezo gating, as currents clearly saturated after leak subtraction (*Figure 1A–D*, bottom). Additionally, we never observed channel openings during the ramp phase of the protocol in 15 patches from Neuro2A-Piezo1ko cells (data not shown). Ultimately, we decided to combine the ramp stimulus protocol with a subsequent brief (200 ms) saturating pressure step to –60 mmHg. Importantly, the step pulse allowed us to confirm post hoc that inactivation of Piezo1 at this voltage is minimal (step pulse: mean current at 200 ms = 96.5±7.4% of peak; n=281 patches; *Figure 1—figure supplement 2A,D*).

Next, we focused on titrating the sizes of our patch pipettes such that a substantial fraction of patches had zero channels (*Figure 1—figure supplement 2E*; see Materials and methods for details). We limited our pipettes to 3–6.5 MΩ, reasoning that this choice maximizes our ability to detect potential channel clusters and thus resolve the overall spatial distribution of channels. Also, to address the possibility that Piezo1 cellular distribution could be altered by the cell-substrate interface, which we reasoned may trap Piezo1 channels and deplete them from cell-attached patch-recordings on the cell roof, we performed day-matched experiments of Neuro2A cells that had either been seeded to adhere overnight or were acutely replated <1 hr before recording, therefore allowing limited time for channel redistribution (see Materials and methods). Importantly, we found that the mean channel number and distributions in these two data sets were identical (mean ± SEM: overnight=2.4±0.4 channels, n=28 patches; acute=2.7±0.4 channels, n=26 patches, p=0.6, Student's t-test; *Figure 1—figure supplement 2F*), which argues against an effect of channel redistribution to the cell-substrate interface. Finally, in order to obtain good representations of the spatial distribution and function of Piezo1 channels, we executed this protocol on patches from a very large number of individual Neuro2A cells (n=281).

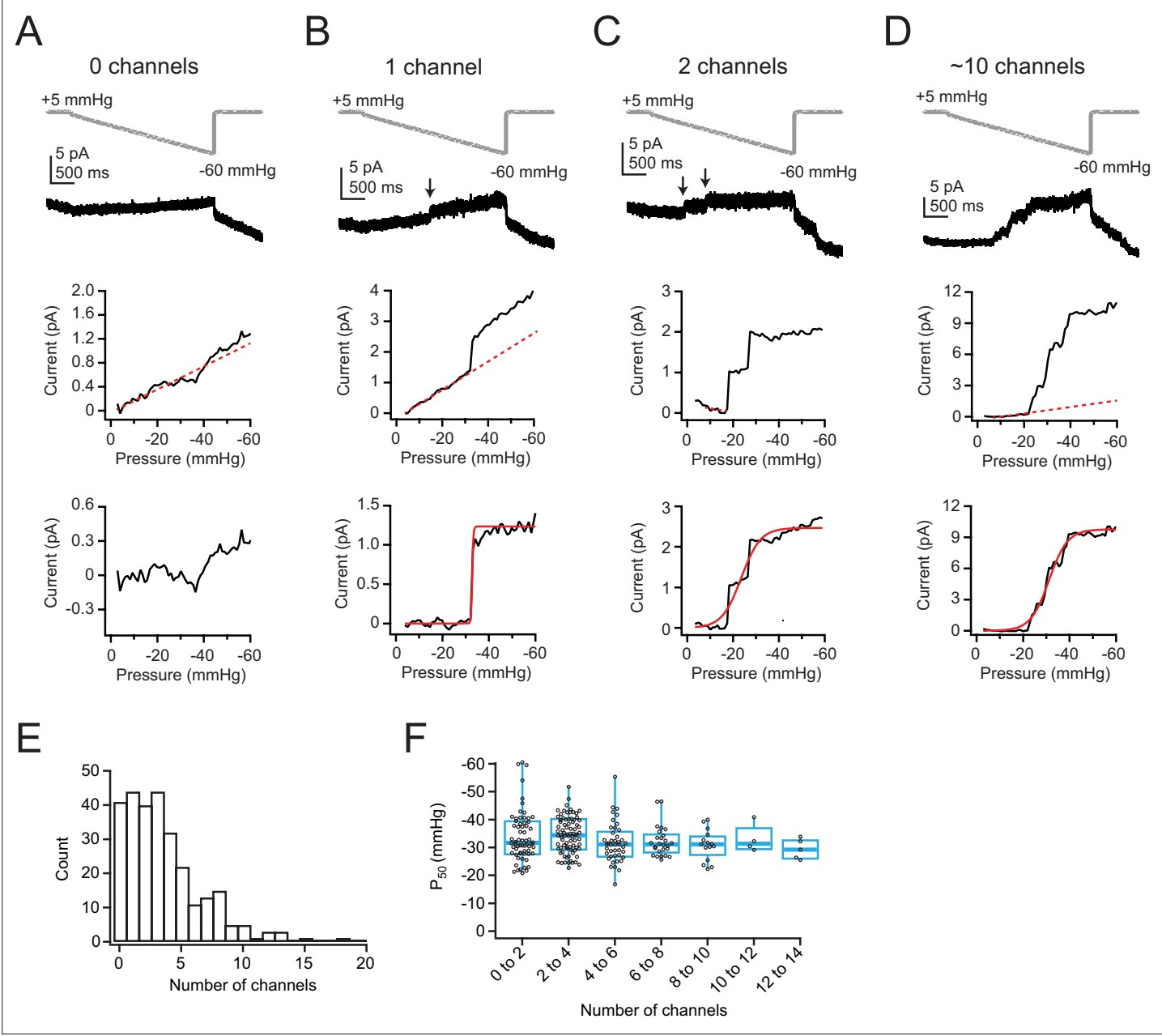

**Figure 1.** Low levels of Piezo1 expression in Neuro2A cells do not influence pressure sensitivity. (**A**) *Top*, Pressure protocol (gray) and current (black) from a cell-attached patch from Neuro2A cells with zero channels. Holding potential was +60 mV. *Middle*, Current-pressure relationship for the same patch. Mean current was calculated for each step, in 1 mmHg increments, and plotted as a function of pressure. Note the linear relationship between current and pressure during the step, despite no channel activity in this patch (inferred from the lack of discrete channel opening events). This is likely due to small changes in seal quality and/or capacitance. The linear portion of the current-pressure relationship was fitted with a line (red dashes). *Bottom*, Current as a function of pressure after subtraction of the current corresponding to the fitted line. (**B**) As in (**A**), for a patch with one channel. Arrow indicates discrete channel opening event. *Bottom*, current was fitted with a Boltzmann equation (see Materials and methods). From the fit (red line), $I_{max}$, which corresponds to single-channel current, was 1.2 pA and $P_{50}$ was –32.8 mmHg. (**C**) As in (**A, B**), for a patch with two channels. From the fit (red line), $I_{max}$ was 2.5 pA and $P_{50}$ was –23.6 mmHg. (**D**) As in (**A–C**), for a patch with many channels. From the fit (red line), $I_{max}$ was 9.8 pA and $P_{50}$ was –31.2 mmHg. (**E**) Histogram of channel number in each patch. n=281 patches. (**F**) $P_{50}$ values, measured from sigmoidal fits to currents in (**A–D**), as a function of channel number. Bin width is two channels. n=281 patches.

The online version of this article includes the following figure supplement(s) for figure 1:

**Source data 1.** Low levels of Piezo1 expression in Neuro2A cells do not influence pressure sensitivity.

**Figure supplement 1.** HUVECs express low levels of Piezo1.

*Figure 1 continued on next page*

*Figure 1 continued*

**Figure supplement 1—source data 1.** HUVECs express low levels of Piezo1.

**Figure supplement 2.** Ramp-like protocol effectively measures $I_{max}$ in Neuro2A cells.

**Figure supplement 2—source data 1.** Ramp-like protocol effectively measures $I_{max}$ in Neuro2A cells.

After linear leak subtraction, we could easily recognize in most patches (248/281) discrete, nearly instantaneous increases in current amplitude that are reflective of individual channel openings, as well as clearly identify a plateau phase of the current (*Figure 1B*, middle). The remaining patches (33/281) were classified as having zero channels (e.g., *Figure 1A*). We next used the mean single-channel current from all patches with one channel (i=0.98±0.04 pA, n=35 patches; *Figure 1—figure supplement 2G*) to calculate the precise number of channels (n=$I_{max}$/i) in all other patches. Ultimately, we obtained a continuous, bell-shaped distribution of Piezo1 channel numbers per patch (*Figure 1E*). The distribution average was 3.5±3.1 channels per patch, which given our small pipette sizes is consistent with previously reported levels of expression in Neuro2A cells (*Coste et al., 2010*). We also applied each protocol twice per patch and then performed pairwise comparison of current amplitudes (*Figure 1—figure supplement 2H-I*). This analysis revealed an extremely low variability (ratio of 2nd/1st recording=1.1±0.4; n=261), supporting that the assessment of channel numbers is highly accurate. We estimated the dome size of membrane patches inside patch pipettes from imaging experiments to be 2 µm (see Materials and methods) and used this value to calculate the average density of Piezo1 channels to be 1–2 channels/µm$^2$.

Next, we used two different approaches to assess if the density of Piezo1 channels affects their sensitivity to membrane tension: First, we calculated $P_{50}$ values, measured from sigmoidal fits to the ramp phase of the current, and plotted them as a function of channel number (*Figure 1F*). Although $P_{50}$ measurements varied widely in individual patches, especially for those with <5 channels, because of the stochastic nature of channel gating (e.g., *Figure 1C*), this variance is partially overcome by the large number of measurements we obtained (n=248). Importantly, while there was some variability between two protocols executed on the same patch, there was only a slight systematic shift in $P_{50}$ values toward smaller pressures in the second protocol, likely due to creep and corresponding increase in patch radius (mean ± SD first=–33.1±7.2 mmHg, second=–31.1±7.7 mmHg; $\Delta P_{50}$=+2.0±8.2 mmHg; p<0.05, paired t-test; *Figure 1—figure supplement 2J*), providing further evidence that our protocol allowed for highly reproducible measurements. Altogether, the results show that, for patches containing 1–14 channels, average $P_{50}$ values do not vary substantially with channel number (*Figure 1F*).

Still, to measure $P_{50}$ values in patches with few channels more precisely, we also performed a different analysis: Specifically, we idealized recordings with ≤5 channels by identifying discrete openings (*Figure 2A–E*; see Materials and methods) and then averaged all idealized traces with equal channel numbers in order to obtain mean pressure-response curves (*Figure 2B*). Visual inspection as well as fits with sigmoidal functions show that $P_{50}$ and slope (k) values are virtually identical for patches containing 1, 2, 3, 4, or 5 channels (1 channel: $P_{50}$=–23.2 mmHg, k=–3.3 mmHg; 5 channels: $P_{50}$=–23.7 mmHg, k=–4.2 mmHg). We therefore conclude from both analyses that at densities of up to 2 channels/µm$^2$, Piezo1 channels do not influence each other's mechanosensitivity.

## Spatial distribution of Piezo1

Multiple mechanisms may explain our above result that Piezo1 pressure sensitivity does not vary with channel density. First, Piezo1 channels may be randomly distributed and therefore spatially too distant to functionally interact via their membrane footprints. Alternatively, Piezo1 channels may indeed tend to localize in close spatial proximity (i.e., groups of 2–3 channels), but this localization may have no effect on their pressure sensitivity. To begin distinguishing among these possibilities, we built a statistical model for spatial localization, based on a Thomas point process (see Materials and methods for details). First, we simulated a distribution of randomly dispersed Piezo1 channels. Then we randomly sampled from this population, accounting for variability of pipette sizes and single-channel currents by using our experimental mean values and standard deviations of pipette size and single-channel conductance (*Figure 1—figure supplement 2E,G*). The resulting distribution fits the experimental

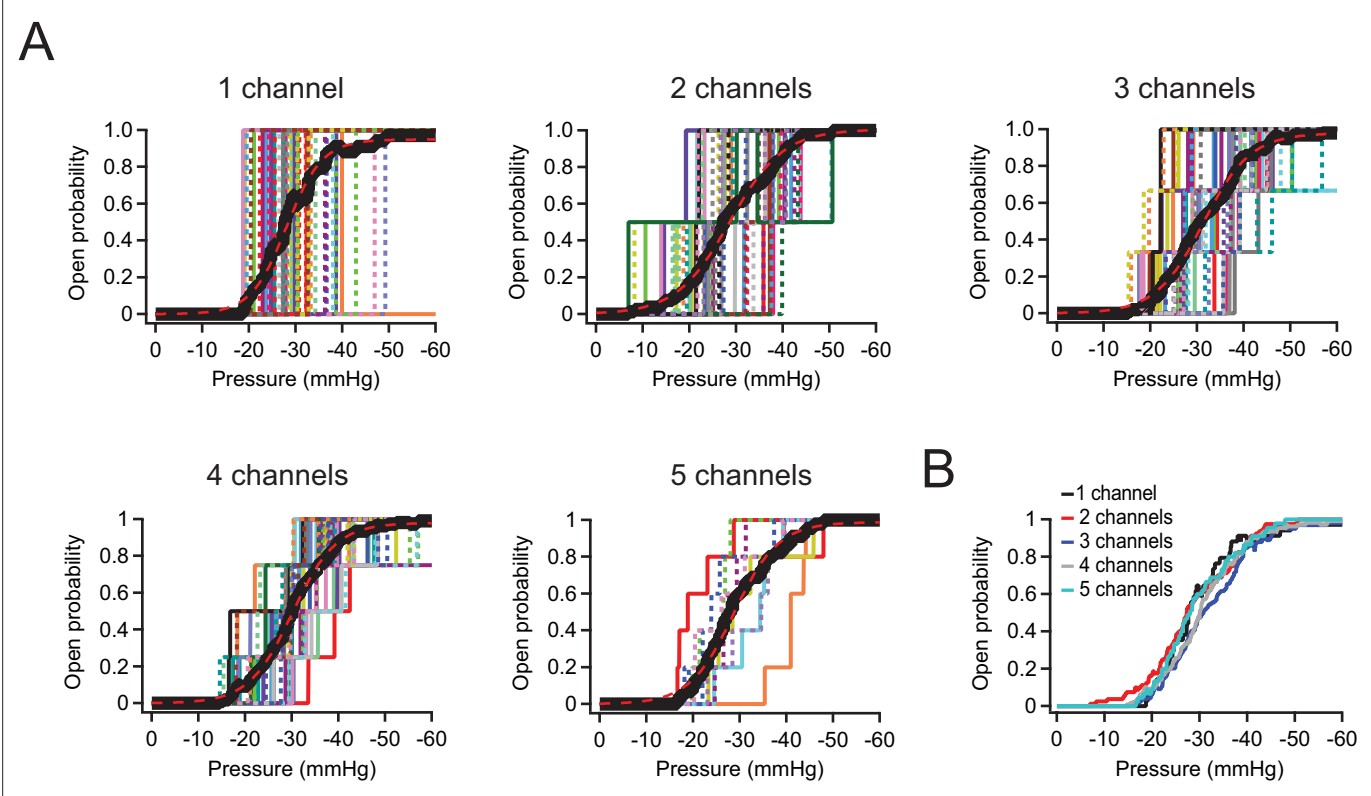

**Figure 2.** $P_{50}$ and slope values are identical for Neuro2A patches with 1–5 Piezo1 channels. (**A**) Idealized current-pressure relationship for all patches with precisely 1–5 channels. Each patch is a different color/weight; thick black line is the mean open probability for all patches (see Materials and methods). Mean open probability was fit with a sigmoidal relationship (dashed red lines): 1 channel: $P_{50}=-23.2$ mmHg, $k=-3.3$ mmHg, n=35 patches. 2 channels: $P_{50}=-23.6$ mmHg, $k=-5.1$ mmHg, n=40 patches. 3 channels: $P_{50}=-25.9$ mmHg, $k=-4.5$ print mmHg, n=33 patches. 4 channels: $P_{50}=-24.9$ mmHg, $k=-4.0$ mmHg, n=27 patches. 5 channels: $P_{50}=-23.7$ mmHg, $k=-4.2$ mmHg, n=9 patches. (**B**) Overlaid mean open probability as a function of pressure for patches with precisely 1, 2, 3, 4, or 5 channels.

The online version of this article includes the following figure supplement(s) for figure 2:

**Source data 1.** $P_{50}$ and slope values are identical for Neuro2A patches with 1–5 Piezo1 channels.

data reasonably well, indicating that in Neuro2A cells, Piezo1 channels may be homogenously distributed across the cell surface, rather than being spatially localized in close proximity (*Figure 3A–F*).

In order to challenge this interpretation, we next simulated a random distribution of Piezo1 clusters in which individual Piezo channels are distributed across a 2D-Gaussian distribution with a standard deviation of 50 nm, which is the estimated size of the Piezo channel footprint (*Haselwandter and MacKinnon, 2018*; see Materials and methods), all while holding the overall channel density constant. Strikingly, a distribution in which on average 1.6 Piezo1 channels (drawn from a Poisson distribution centered at 1; see Materials and methods) interact over 50 nm also describes the experimental data well. However, simulating larger channel clusters (n ~ 5) fails to reproduce the experimental distribution, specifically the probability of capturing patches without any channels (*Figure 3A–F*). Thus, our experimental data are most parsimoniously explained by either a random spatial distribution or a weak propensity for Piezo channels to spatially localize in groups of 2–3. Interestingly, the two distributions differ significantly in the predicted fraction of channels that are within one footprint of at least one other channel: in a truly random distribution, the mean nearest-neighbor distance is ~390 nm, and only ~ 5% of channels are within 100 nm of another channel, whereas if channels tend to cluster in groups of 2–3, the mean nearest-neighbor distance is reduced to 225 nm and almost 50% of channels are within 100 nm of another channel (*Figure 3G–H*). However, our functional data support the idea that if the latter spatial localization indeed occurs, it does not have a substantial effect on Piezo1 pressure sensitivity (*Figures 1 and 2*).

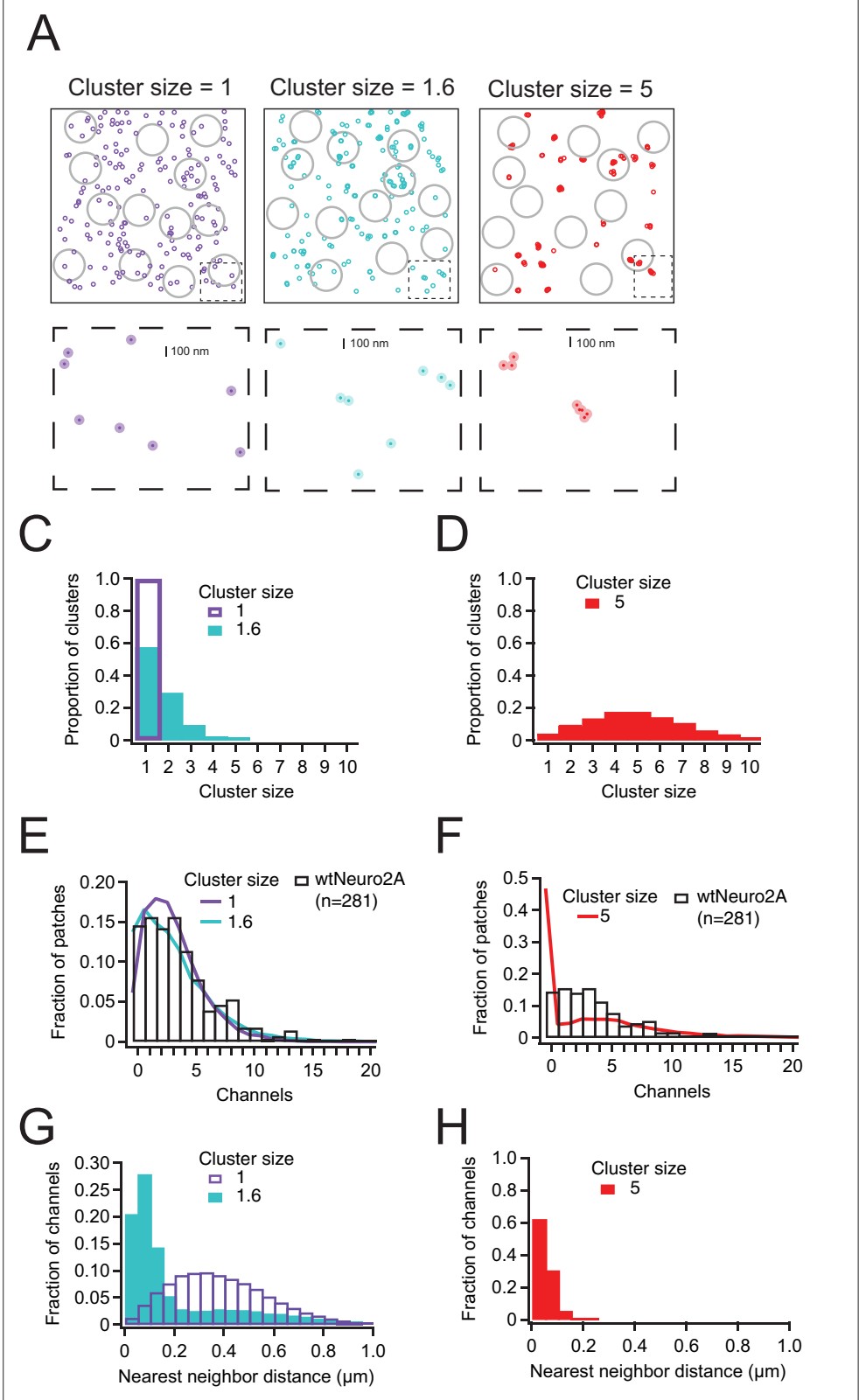

**Figure 3.** Piezo1 distribution in Neuro2a cells is best explained by little or no clustering of channels. (**A**) Representative channel distributions generated using a Thomas point process with overall densities equivalent to that of wild-type Neuro2A cells (~1.75 channels/µm²); in each condition, daughter 'channels' are assigned to a center 'parent' disc, with a mean distance of 50 nm (~1 Piezo footprint) from the center of the disc. Three separate

*Figure 3 continued*

degrees of clustering (cluster size) were introduced; one in which every cluster has precisely one channel, one with a mean of 1.6 channels per cluster, and one with a mean of 5 channels per cluster. Gray circles represent model patch domes used to sample the distributions, with a mean radius of 0.8 µm. Below, insets (dashed boxes) of each distribution; dark circles are the projected area of the Piezo footprint to scale; light circles are ~ 3× the membrane footprint for each channel (radius=50 nm). (**C, D**) Distributions of channels per cluster for mean cluster sizes of 1 and 1.6 (**C**) and 5 (**D**). (**E, F**) Normalized histogram of channels per patch for wild-type Neuro2A cells for mean cluster sizes of 1 and 1.6 (**E**) and 5 (**F**). (**G, H**) Histogram of nearest-neighbor distances for mean cluster sizes of 1 and 1.6 (**G**) and 5 (**H**).

The online version of this article includes the following figure supplement(s) for figure 3:

**Source data 1.** Piezo1 distribution in Neuro2a cells is best explained by little or no clustering of channels.

## Piezo1 function at high channel densities

Our above results do not answer if, in principle, Piezo1 channels have the ability to functionally interact. We therefore decided next to examine Piezo1 function using a heterologous overexpression system, in which channel densities are much higher. For these experiments, we chose Neuro2A-Piezo1ko cells, in order to retain the same cellular background, but avoid heterogeneity from endogenous and over-expressed Piezo1, especially given that there are known splice variants of Piezo1 (*Geng et al., 2020*). In pilot experiments, we found that transfection with Lipofectamine 2000 maximizes channel density 42–48 hr post-transfection (data not shown). We first repeated our previous ramp protocol, but found that while overexpression yields peak currents that are much larger, these currents inactivate to a larger extent even at +60 mV, such that a ramp protocol substantially underestimates the total number of channels in the patch (*Figure 4—figure supplement 1*; current at 200 ms=53.3±18.1% of peak, n=138 patches). Therefore, we instead chose to use a classic pressure-step protocol to best measure both peak current ($I_{max}$) and pressure sensitivity ($P_{50}$ and k) (*Figure 4A–B*). While the step protocol still produces a 2–3 pA background current at high pressures, owing to changes in leak and/or capacitance, here, this error is very minor compared to the large Piezo-mediated currents (generally 20–200 pA), especially for patches with many channels. To reduce bias from resting membrane tension, each pressure step was preceded by a 5 s +5 mmHg prepulse, the first 4 s of which was applied at –80 mV to minimize the cumulative effects of positive voltage on membrane creep (*Suchyna et al., 2009*). We used a pressure increment (ΔP) of –5 mmHg, so that we could better resolve small changes in pressure sensitivity, but limited the duration of each step to 250 ms to again minimize membrane creep (*Figure 4A and B*).

The low channel activity during the prepulse allowed us to measure the single-channel current of each individual patch at both –80 and +60 mV (*Figure 4—figure supplement 2B,C*) and therefore to precisely calculate the minimal number of channels ($n=I_{max}/i$). This is particularly important in a cell-attached configuration, where even in a high $K^+$ solution, there can be slight variance in the resting membrane potential and correspondingly variance in the single-channel current in each cell that introduces an additional source of error. Importantly, there was no correlation between number of channels and single-channel current, suggesting that unitary conductance of Piezo1 is not modulated by channel density (*Figure 4—figure supplement 2E,F*). As before, we restricted our pipettes to 3–6.5 MΩ, a range in which peak current amplitudes do not vary substantially (*Figure 4—figure supplement 3A*). We again repeated this measurement for many individual patches (n=144 patches) in order to capture an accurate representation of channel densities and functional interactions.

The number of channels per patch was well described by a Gaussian distribution with 89.2±49.8 channels per patch (*Figure 4C*), which we estimate corresponds to a Piezo1 density of ~45 channels/µm² (see Materials and methods). The most extreme current amplitude we observed (–190 pA, or ~200 channels) corresponds to a maximal density of ~100 channels/µm². At this maximal density, if Piezo channels were randomly distributed in the membrane, their mean next-neighbor distance would be ~50 nm (*Figure 4—figure supplement 4*). This calculation suggests that we have reached densities in which Piezos are likely in close enough proximity to influence each other's function. Moreover, this calculation likely represents a slight underestimate of the true channel density, as residual inactivation at positive potentials may lead to a population of 'silent' channels that do not manifest themselves functionally.

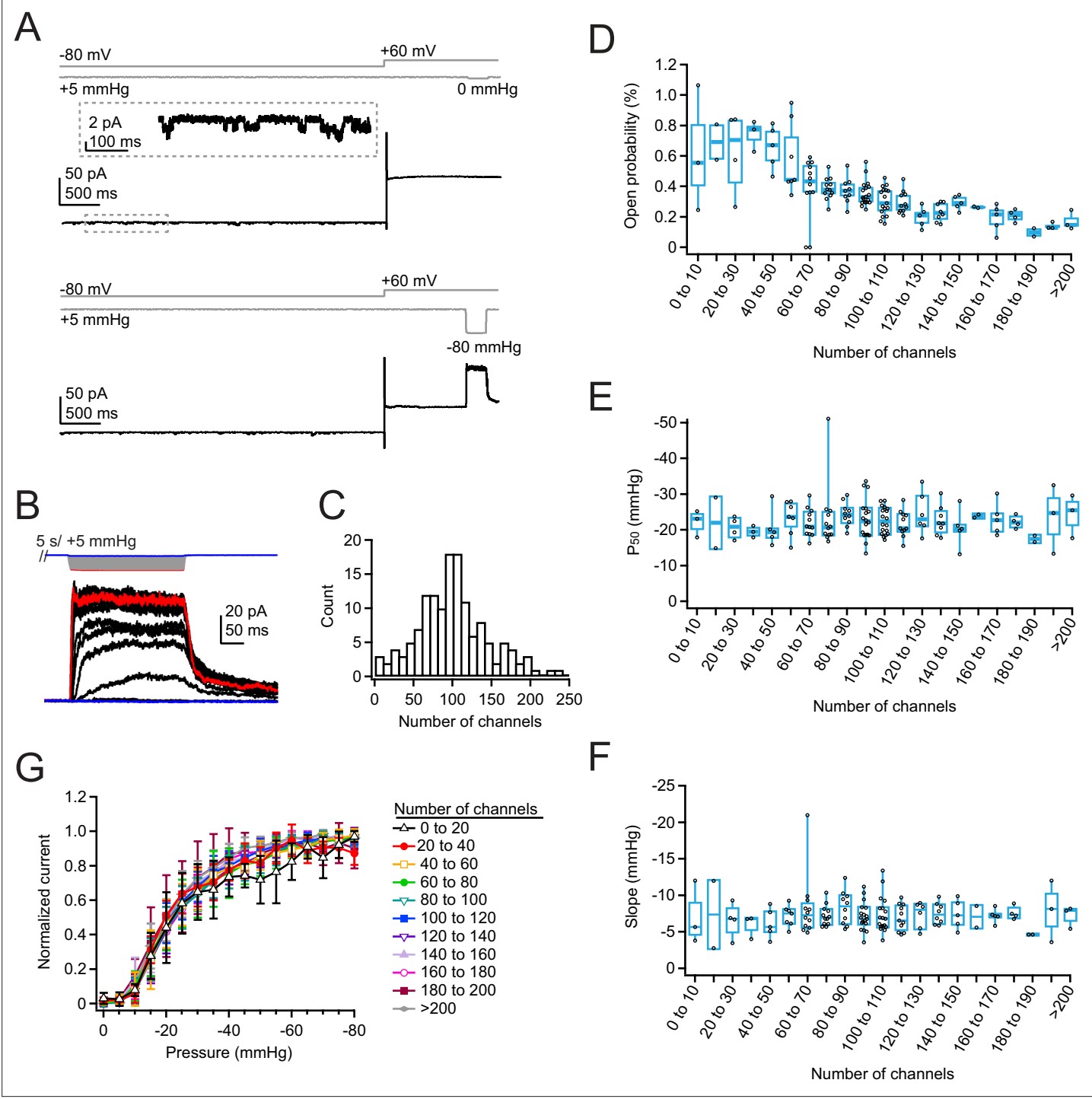

**Figure 4.** Density of overexpressed Piezo1 channels does not affect resting open probability or pressure sensitivity. (**A**) Pressure protocol (gray) and current from a cell-attached patch from a Neuro2A-p1ko cell overexpressing mouse Piezo1. Pressure steps to 0 mmHg (top) and –80 mmHg (bottom) were preceded by a 5 s prepulse to +5 mmHg to relieve resting membrane tension. Top inset shows spontaneous single-channel activity at –80 mV during the prepulse, used to calculate open probability in the nominal absence of tension. (**B**) Full pressure-response protocol and currents for the cell in (**A**). Pressure steps were from 0 to –80 mmHg in –5 mmHg increments. (**C**) Histogram of channel counts, obtained by dividing the peak current elicited during the protocol in (**A**) by the single-channel current measured from unitary events (see also Figure S5). n=144 patches. (**D**) Open probability during the +5 mmHg prepulse as a function of number of channels. n=2–18 patches. (**E**) $P_{50}$ values measured from sigmoidal fits to current-pressure relationships from the protocol in (**B**) as a function of number of channels. (**F**) Slope (k) values measured from sigmoidal fits to current-pressure relationships from the protocol in (**B**) as a function of the number of channels. (**G**) Mean pressure-response curves, generated by first normalizing each

*Figure 4 continued on next page*

*Figure 4 continued*

patch to its maximal current, binning by number of channels, then averaging normalized currents at each pressure. n=2–18 patches.

The online version of this article includes the following figure supplement(s) for figure 4:

**Source data 1.** Density of overexpressed Piezo1 channels does not affect resting open probability or pressure sensitivity.

**Figure supplement 1.** Ramp-like protocol induces substantial inactivation in overexpressed Piezo1 channels.

**Figure supplement 1—source data 1.** Ramp-like protocol induces substantial inactivation in overexpressed Piezo1 channels.

**Figure supplement 2.** Single-channel current measurements from overexpressed Piezo1.

**Figure supplement 2—source data 1.** Single-channel current measurements from overexpressed Piezo1.

**Figure supplement 3.** Pipette resistance does not affect Piezo1 open probability or pressure sensitivity.

**Figure supplement 3—source data 1.** Pipette resistance does not affect Piezo1 open probability or pressure sensitivity.

**Figure supplement 4.** Estimation of nearest-neighbor distances for overexpressed Mouse Piezo1.

**Figure supplement 4—source data 1.** Estimation of nearest-neighbor distances for overexpressed Mouse Piezo1.

Two groups had previously predicted that nearby Piezo1 channels may influence each other's gating, even in the absence of membrane tension (*Haselwandter and MacKinnon, 2018*; *Jiang et al., 2021*). We therefore chose first to analyze the activity during the prepulse, which is designed to minimize resting membrane tension. We found that open probability was generally small (<1%), which was consistent with previous work in our lab (*Wu et al., 2016*). Most notably however, we found that Piezo1 open probability did not increase with channel density; rather, open probability slightly decreased, from 0.6±0.4% for patches with 1–10 channels (mean ± SD; n=3 patches) to 0.2±0.1% for patches with >200 channels (n=3 patches) (*Figure 4D*). There was no change in open probability with pipette resistance, indicating that variability in the ability of the prepulse to minimize resting tension with pipette size does not account for this trend in open probability with channel number (*Figure 4— figure supplement 3B*).

As an alternative test, we measured the effect of channel number on pressure sensitivity. Both slope and $P_{50}$ values were unaffected by channel number (*Figure 4E–F*), and these values were also unaffected by pipette size (*Figure 4—figure supplement 3C-D*). However, patches with small ensembles of channels (<20) yielded imprecise estimates of $P_{50}$ and slope values for two reasons: First, for patches with few channels, the stochastic nature of channel gating generates highly variable individual pressure-response curves. Second, the relatively few patches we obtained with low channel numbers resulted in large errors in mean value estimates. Therefore, we also analyzed the data in a separate way: we normalized each patch to its maximal current, binned patches according to their number of channels, and then averaged pressure-response curves (*Figure 4G*). The results again show that pressure-response curves and sensitivity (slope) are independent of channel numbers (for 0–20 channels, $P_{50}$=–21.5 mmHg, k=–8.0 mmHg, n=5; for >200 channels, $P_{50}$=–21.0 mmHg, k=–6.8 mmHg, n=3).

Finally, we combined our electrophysiological pressure-step protocol with differential interference contrast (DIC) imaging to directly measure the relationship between channel density and membrane tension (*Lewis and Grandl, 2015*; *Figure 5A–C*). In these experiments, we were able to independently calculate channel density for each individual patch by dividing the number of channels in each patch by its visualized surface area at 0 mmHg (see Materials and methods; *Figure 5—figure supplement 1*). This more detailed approach also allowed us to rule out that deviations in pipette size create a systematic bias on $P_{50}$ values, as radius, and therefore tension, is measured for each individual patch and applied pressure. Importantly, we found no correlation between either the tension of half-maximal activation ($T_{50}$) or the slope (k) and channel density in the range of 10–70 channels/μm$^2$ (*Figure 5D–F*), again supporting that Piezo1 channels do not influence each other's tension sensitivity. Altogether, we conclude that at low and high densities, Piezo1 channels show no positive cooperativity in the presence or in the nominal absence of membrane tension.

## Discussion

We sought to characterize the inherent ability of Piezo channels to spatially localize and modulate each other's function. We chose cell-attached patch-clamp electrophysiology as a method for investigation, which has several inherent limitations that need to be considered.

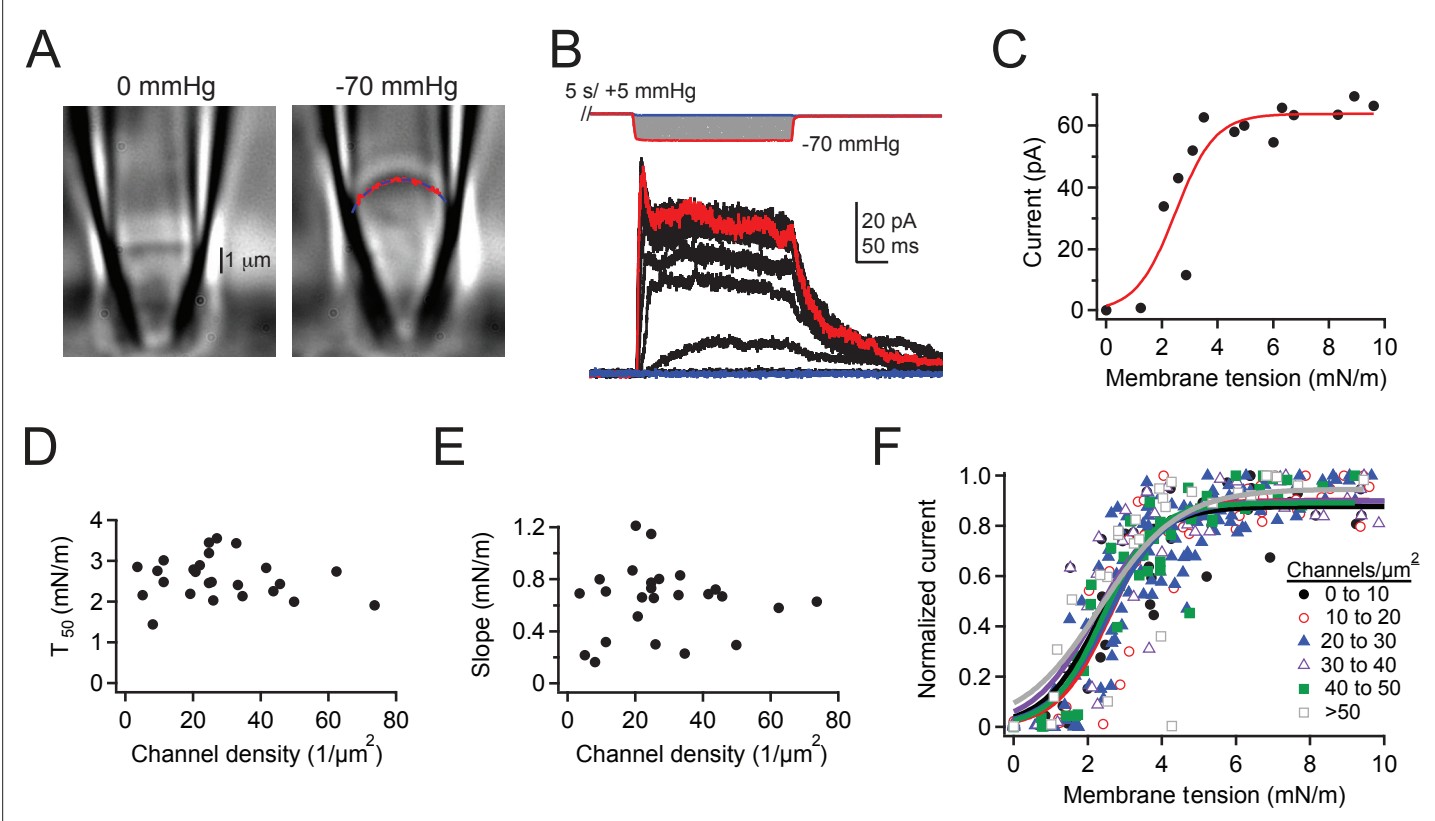

**Figure 5.** Density of overexpressed Piezo1 channels does not affect tension sensitivity. (**A**) DIC image of a cell-attached patch from a Neuro2A-p1ko cell expressing mouse Piezo1 at 0 mmHg (left) and –70 mmHg (right). Red dashed line represents a circle fit to the membrane outline in blue and was used to calculate patch radius (see Materials and methods). (**B**) Full pressure-response protocol and currents from the patch in (**A**). Pressure steps were from 0 to –70 mmHg in –5 mmHg increments. (**C**) Peak current during the pressure step as a function of tension for the patch in (**A**), calculated from patch radius using Laplace's law (see Materials and methods). Red line is a sigmoidal fit used to calculate half-maximal tension for activation ($T_{50}$)and slope (k) using the equation. $I = \frac{I_{max}}{1+e^{\frac{T_{50}-T}{k}}}$ (**D, E**) $T_{50}$ and slope values (k) as a function of channel density. n=25 patches. (**F**) Mean tension-response curves, generated by first normalizing each patch to its maximal current, binning by channel density, then fitting all data as a function of tension. 0–10 channels: $T_{50}$=2.3±0.2 mN/m, k=0.8±0.1 mN/m; 10–20 channels: $T_{50}$=2.6± 0.2 mN/m, k=0.7±0.2 mN/m; 20–30 channels: $T_{50}$=2.6 ± 0.1 mN/m, k=0.8 ± 0.1 mN/m; 30–40 channels: $T_{50}$=2.2±0.2 mN/m, k=0.9 ± 0.2 mN/m; 40–50 channels: $T_{50}$=2.4 ± 0.1 mN/m, k=0.7 ± 0.1 mN/m; >50 channels: $T_{50}$=2.3±0.4 mN/m, k=1.1±0.4 mN/m. n=2–9 patches. DIC, differential interference contrast.

The online version of this article includes the following figure supplement(s) for figure 5:

**Source data 1.** Density of overexpressed Piezo1 channels does not affect tension sensitivity.

**Figure supplement 1.** Estimation of patch dome size as a function of pipette resistance.

**Figure supplement 1—source data 1.** Estimation of patch dome size as a function of pipette resistance.

First, our spatial resolution is naturally limited by the size of the patch pipette. Consequently, our statistical approach of counting channels within membrane patches works best when the spatial distance between Piezo1 channels is comparable to the size of the membrane dome inside the patch pipette. The fact that ~ 15% of patches in wild-type Neuro2A cells contained no Piezo1 channels (*Figure 1E*) is therefore an extremely important quality benchmark for adequate spatial resolution. Our modeling does not allow us to rule out the possibility of moderate clustering (2–3 Piezo1 channels), but we find no evidence for clusters of larger sizes (~5 channels), which we would have been able to detect (*Figure 3*).

Next, our measurement of endogenous Piezo1 density in Neuro2A cells (1.75 channels/µm²) has several sources of error: first, while we have modest control over the pipette shape during its fabrication and a precise measure of its resistance, without simultaneous DIC imaging, the exact surface of each individual membrane dome is not known. In order to minimize this size variability, we therefore limited the majority of our analyses to measurements from a narrow range of pipette resistances and

incorporated this size variability of patch pipettes into our statistical model of Piezo1 spatial distribution (*Figure 1—figure supplement 2D*). Additionally, we performed a smaller subset of experiments combining simultaneous DIC imaging with electrophysiology (*Figure 5*).

Certainly, when overexpressing Piezo1 the variability in Piezo1 membrane levels is high, in part due to variable expression levels among cells. Assessing Piezo1 expression levels is also subject to error due to the presence of inactivation, which prevents reaching a $P_o$=1. Specifically, channels inactivating during the rising phase of the current, as well as the presence of 'silent' channels that fail to respond to the stimulus entirely, contribute to this error. However, none of the above statistical errors, or others we did not consider and that skew channel number distributions, affect our ability to correlate Piezo1 channel number with its function.

The weaknesses of cell-attached patch-clamp electrophysiology are balanced by two clear strengths: the method has relatively high throughput so that large numbers of independent measurements yield good population averages, and it has unmatched precision: channel density and function, including single-channel current, open probability, and pressure sensitivity can be extracted over 2 orders of magnitude (1–100 channels/$\mu m^2$) with high accuracy. For example, repeated execution of pressure ramps on each patch revealed that channel number (n) is measured with only 24% variability, demonstrating that this assessment is extremely accurate.

Neither endogenous Piezo1 in Neuro2A cells or overexpression of Piezo1 in Neuro2A-P1ko cells are perfect model systems for investigating how Piezo1 channels function when they are directly contacting each other: experimental channel densities do not reach the theoretical limit of a tightly packed 2D hexagonal lattice. However, the maximal densities we obtained are certainly sufficient for Piezo1 membrane footprints to geometrically overlap: for example, at the highest density we measured (~100 channels/$\mu m^2$), the centers of randomly spaced Piezo1 domes are on average 50 nm from their nearest-neighbor, which results in a substantial number of channels having overlapping membrane footprints (*Figure 4—figure supplement 4*; *Haselwandter and MacKinnon, 2018*). In any case, two distinct analyses show that at densities of 1–2 channels/$\mu m^2$ (*Figures 1–2*) and 10–100 channels/$\mu m^2$ (*Figures 4–5*), Piezo1 pressure and tension sensitivity are not modulated by channel numbers. A mechanistic explanation for this result at low densities is directly provided by our data and statistical model of spatial distributions, which suggest that in Neuro2A cells, Piezo1 channels are either randomly distributed or have at best a very low probability of aggregating. This is consistent with Haselwandter and MacKinnon's prediction that the Piezo structure favors channels to repel each other. However, the result does not mean that Piezo1 channels in close proximity can never be found. At native expression levels, a mild tendency for clustering predicts 20% of all Piezo1 channels will be within 50 nm of each other (cluster size=1.6; *Figure 3G*), which is fully consistent with studies that have captured images of neighboring Piezo1 channels (*Ridone et al., 2020*), but 80% of all Piezo1 channels are outside this range. In other words, in wild-type Neuro2A cells, most Piezo1 channels are too far apart to 'feel' each other. As a consequence, they behave as functionally independent mechanotransducers. This result is important, because it shows that Piezo1 channels have no strong intrinsic tendency to spatially concentrate and functionally modulate each other's mechanical sensitivity. Both of these properties are ideal for a cell in order to transduce mechanical forces homogeneously.

Still, does the Piezo1 membrane footprint allow for the possibility of cooperativity among nearby channels? Our experiments at high channel densities (up to 100 channels/$\mu m^2$) reveal no change in Piezo1 pressure sensitivity with increasing channel numbers. Surprisingly, our data do suggest that in the nominal absence of membrane tension Piezo1 open probability is slightly (~0.5%) reduced with channel number but overall, very low (<1%) at all channel densities we probed. This is notable, because the opposite result was recently obtained by simulations of Piezo1 that are in extreme proximity to each other (1–3 nm dome-dome distance; *Jiang et al., 2021*). The reasons for this discrepancy may be technical limitations in the modeling, which was performed with a partial Piezo1 structure, or our own experiments, which cannot generate the extreme channel densities probed in the simulation. Moreover, any cooperativity need not arise via membrane footprint interactions: for example, calcium permeating through Piezo1 may influence the resting open probability of nearby channels. Indeed, calcium-induced cooperativity is a well-known phenomenon in other ion channels, such as TRPA1 (*Wang et al., 2008*). Our experiments cannot directly rule out this explanation.

Altogether, our conclusions may be applicable to other cell types that express Piezos at levels similar or lower to that of Neuro2A cells. However, we fully expect that specific cell types may concentrate

Piezo channels into locations that are dedicated to detecting forces. For example, it is plausible that in sensory neurons Piezo2 channels may be highly concentrated in free nerve endings as compared to the cell soma. However, our work directly predicts that if such a clustering was indeed observed, it would require a dedicated mechanism that is not Piezo-intrinsic, such as a still elusive tethering molecule.

# Materials and methods

**Key resources table**

| Reagent type (species) or resource | Designation | Source or reference | Identifiers | Additional information |
|---|---|---|---|---|
| Cell line (*Mus musculus*) | Neuro2A | ATCC#CCL-131 | | |
| Cell line (*M. musculus*) | Neuro2A-p1ko | Gift of Gary Lewin; *Moroni et al., 2018* | | |
| Cell line (*Homo sapiens*) | HUVECS, pooled | Clonetics CC-2519 | | |
| Transfected construct (*M. musculus*) | Piezo1-IRES-GFP | Gift of Ardem Patapoutian; *Coste et al., 2010* | | |

## Cell culture

Wild-type Neuro2A cells (ATCC #CCL-131) and Neuro2A-p1ko cells (a gift of Gary Lewin) were cultured at 37°C and 5% $CO_2$ in Minimum Essential Medium (Thermo Fisher Scientific) supplemented with 0.1 mM non-essential amino acids, 1 mM pyruvate, 10% fetal bovine serum (Clontech), 50 U/ml penicillin, and 50 mg/ml streptomycin (Life Technologies). Both cell lines were STR authenticated by ATCC and tested in-house for mycoplasma via PCR (ATCC #30-1012K). HUVECs (Clonetics CC-2519, pooled) were cultured in EGM-1. For experiments without transfection, cells were directly plated on poly-L-lysine and laminin-coated coverslips 16–24 hr before recording. For overexpression experiments, cells were transiently transfected with 4 µg mouse Piezo1-pIRES-EGFP in pcDNA3.1(+) or 4 µg YFP 40–48 hr before recording in six-well plates using Lipofectamine 2000 (Thermo Fisher Scientific) according to the manufacturer's protocol. Transfected cells were reseeded onto poly-L-lysine and laminin-coated coverslips or glass-bottomed dishes (for imaging; MatTek Corporation P50G-0- 30F) 16–24 hr before recording. Only one patch was made from each cell; in some cases, two protocols were run on each patch (as noted). For acute plating experiments, cells were reseeded onto poly-L-lysine and laminin-coated coverslips, allowed to settle for 45 min, and then recorded for up to one additional hour.

## Electrophysiology

Electrophysiological recordings were performed at room temperature using an EPC10 amplifier and Patchmaster software (HEKA Elektronik, Lambrecht, Germany). Data were sampled at 5 kHz (wild-type Neuro2A) or 10 kHz (Neuro2a-p1ko + mPiezo1) and filtered at 2.9 kHz. The cell-attached bath solution used to zero the membrane potential was (in mM): 140 KCl, 10 HEPES, 1 $MgCl_2$, 10 glucose, and pH 7.3 with KOH. Borosilicate glass pipettes (1.5 OD, 0.85 ID, Sutter Instrument Company, Novato, CA) were filled with pipette buffer solution (in mM): 130 NaCl, 5 KCl, 10 HEPES, 10 TEACl, 1 $CaCl_2$, 1 $MgCl_2$, and pH 7.3 with NaOH. In a small dataset obtained with pipettes with a resistance of less than 3 MΩ, peak current was strongly correlated with pipette size and few patches had zero channels (*Figure 1—figure supplement 2E*); we therefore restricted our final data set to pipettes between 3 and 6.5 MΩ. Negative pressure was applied through the patch pipette with an amplifier-controlled high-speed pressure clamp system (HSPC-1; ALA Scientific Instruments, Farmingdale, NY). All pressure protocols were preceded by a +5 mmHg prepulse (ramp protocol, 2 s at +60 mV; step protocol, 4 s at –80 mV, and 1 s at +60 mV) to remove resting tension due to the gigaseal (*Lewis and Grandl, 2015*).

## Imaging

Imaging was performed as described (*Lewis and Grandl, 2015*). Briefly, images were captured at a rate of ~13 frames/s at a resolution of 61.5 pixels/µm using a Plan Apo ( 100×) DIC oil objective coupled with a Coolsnap ES camera and 4× relay lens (Nikon Instruments). Images were analyzed

in Igor Pro 8.02 (WaveMetrics) using custom scripts available in our Github repository (github.com/GrandlLab). The membrane was identified by performing a line scan parallel to the pipette walls and localizing the minimum pixel intensity over a rolling average of 5–9 pixels and fitting the output with a circle to obtain radius (R). Tension (T) was then calculated for each pressure step (p) using Laplace's law: T=R*p/2. The surface area of each patch (*Figure 5—figure supplement 1*) was calculated from the radius at 0 mmHg by fitting the patch dome as a spherical cap and manually estimating the contact points between the membrane and the pipette walls.

## Quantification and statistical analysis

All data were analyzed and final plots were generated using Igor Pro 8.02 (WaveMetrics).

For currents measured during ramp protocols, current was binned for each pressure (1 mmHg increments) and any linear changes in current prior to the first channel opening were manually identified, attributed to capacitive and/or leak changes, and subtracted off prior to further analysis. Current-pressure curves were fit with a Boltzmann function

$$I = I_{min} + \frac{I_{max}}{1 + e^{\left(\frac{P50 - P}{k}\right)}}$$

where $I_{max}$ is the maximal current, P is pressure, $P_{50}$ is the pressure of half-maximal activation, and k is the slope factor. The number of channels in each patch was calculated by dividing $I_{max}$ by the mean single-channel current from all patches with precisely one channel (i.e., $I_{max}$ during the ramp phase for patches with only one visually identified discrete opening, *Figure 1*) or from single-channel openings during the +60 mV prepulse (step recordings with overexpressed channels, *Figures 4 and 5*). In some recordings, no unitary events were observed at +60 mV during the +5 mmHg prepulse, and pressure was manually adjusted in 1 mmHg increments until single-channel openings were elicited.

Single-channel currents during the +5 mmHg prepulse were filtered offline (1 kHz) using Fitmaster v2 × 90.2 and baseline-subtracted using QuB online software (*Nicolai and Sachs, 2014*). Single-channel amplitudes during the prepulse were measured by generating all-points histograms with binning calculated using the Freedman-Diaconis method and an optimal bin width of $2*IQR(x)/N^{1/3}$, where IQR is the interquartile distance, N is the number of observations, and the bins are evenly distributed between the minimum and maximum values. Binned data were then fit with a double-Gaussian equation of the form

$$y = y_0 + A_1 * e^{-\left(\frac{x - x_1}{w_1}\right)^2} + A_2 * e^{-\left(\frac{x - x_2}{w_2}\right)^2}$$

where $y_0$ is the baseline current, $A_1$ and $A_2$ are the peak amplitudes, $x_1$ and $x_2$ are the centers of the fits, and $w_1$ and $w_2$ are their respective widths. The difference between $x_1$ and $x_2$ reflects the difference between the mean current in the open and closed state and was used to calculate single-channel current i.

Open probability during the prepulse was calculated as the mean current ($i*n*P_o$) divided by the maximum current elicited for that patch at saturating pressures, assumed to be $P_o$=1 ($i*n$). For currents measured during step protocols, to reduce artifacts from noise, the peak was measured from a 0.5 ms rolling average around the true peak and the steady-state current was calculated as the mean current during the last 10 ms of the step. To generate idealized currents (*Figure 2*), the number of channels (n) in each patch, as well as the time of openings, were manually identified. The open probability at a given time was taken as the fraction of total channels in the patch that were open. Owing to the lack of inactivation of endogenous currents at +60 mV, as well as the continually increasing pressure, channel closings were rare. Patches with larger numbers of channels (3–5) are underrepresented due to the increased difficulty in identifying discrete openings. Data are reported as mean ± SD unless otherwise indicated. For box and whisker plots, boxes represent median and first and third quartiles.

## Thomas clustering model

A model for spatial distribution of Piezo1 ion channels was built using a Thomas point process in IGOR Pro 8.02 (WaveMetrics) using custom scripts available in our Github repository (https://github.com/GrandlLab/PiezoClusteringModel copy archived at swh:1:rev:c4463867f753c41d4a92f-f25a03ceb6131bfbc22, *Baddeley, 2015*, *Grandl, 2021*). Parent discs were randomly distributed in

a 50 µm × 50 µm arena. Each parent disc was then randomly assigned a varying number of daughter points. For a truly random distribution (mean cluster=1) every parent disc was assigned precisely one daughter point. To introduce a slight propensity for channels to cluster, the number of daughter points were drawn from a Poisson distribution centered at 1, such that the mean cluster size (for populated clusters) was 1.6. To introduce a stronger propensity to cluster, the number of daughter points was drawn from a Poisson distribution centered around 5 (mean cluster=5); *Figure 3A*. The densities of parent discs and daughter points were adjusted such that their product, and thus the final number of daughter points in the arena, was equivalent to that of the wild-type Neuro2A data set (1.75 channels/µm²). Daughter points were then distributed in each parental disc according to a Gaussian distribution with an average distance of sigma=50 nm from the center of the disc, or roughly the distance of three Piezo footprint decay lengths. The arena was then sampled with 1000 circles whose centers were uniformly distributed within the arena. To replicate variability in pipette size/patch dome size as source of noise, the mean radius of all circles was 0.8 µm (see below) with a standard deviation of 0.15 µm, which corresponds to our mean pipette size of 4.35 MΩ and standard deviation of 0.8 MΩ (*Figure 5— figure supplement 1*). Finally, the number of daughter points (channel counts) within each circle was then counted. To further account for variability in transmembrane potentials as a source of noise, we translated channel counts into current amplitudes using our experimentally obtained normal distribution of single-channel currents, which yielded a mean single-channel current of 0.98 pA and standard deviation of 0.22 pA (*Figure 1—figure supplement 2G*). Current amplitudes were then reverted into channel counts and used to generate a channel count histogram. Nearest-neighbor distributions were obtained from the same simulated distributions as above, by calculating a distance matrix between all channel centers and taking the minimal distance for each row.

## Calculation of channel footprint and density

The characteristic decay length ($\lambda$) of membrane deformation induced by the large curved dome of Piezo1 is estimated to be 14 nm (*Haselwandter and MacKinnon, 2018*). We put upper bounds of possible Piezo energetic interactions due to membrane curvature beyond the dome itself as 3× this decay length; when adding this to the radius of the channel dome, this yields an approximate total membrane footprint of (10 nm +3*14 nm ~ 50 nm).

From our imaging data (*Figure 5* and *Figure 5—figure supplement 1*), we estimated that patch domes from pipettes 3 to 6 MΩ have an approximate surface area of 2 µm². If the patch dome is approximated as a flat circle, this corresponds to a pipette radius of 0.8 µm, which was used for simulation of Piezo1 spatial distribution in the Thomas point process model. For native expression in Neuro2A cells, our average channel count was 3.5 channels, which corresponds to an average density of 1.75 channels/µm². For overexpression of Piezo1 in Neuro2A-p1ko cells, our average channel count was ~100 channels, which corresponds to an average density of 50 channels/µm². For the maximal channel number we observed (~200 channels), with an estimated in-plane area of 400 nm² (*Wang et al., 2019*), an estimated footprint of 7800 nm² (calculated from a footprint of 50 nm; *Haselwandter and MacKinnon, 2018*) and a patch dome surface area of 2 µm², approximately 4% of the membrane surface would be covered by Piezo channels and 78% of the surface covered by Piezos + corresponding footprints if channels were evenly distributed (with no overlap of footprints).

## Acknowledgements

This study was supported by NIH 5R01NS110552 (AHL and JG) and Duke University. The authors thank Gary Lewin for generously providing Neuro2A-P1ko cells, and all members of the Grandl Lab for thoughtful comments on the study.

## Additional information

### Funding

| Funder | Grant reference number | Author |
|---|---|---|
| National Institute of Neurological Disorders and Stroke | 5R01NS110552 | Jorg Grandl |

The funders had no role in study design, data collection and interpretation, or the decision to submit the work for publication.

### Author contributions

Amanda H Lewis, Conceptualization, Data curation, Formal analysis, Investigation, Methodology, Software, Writing – original draft, Writing – review and editing; Jörg Grandl, Conceptualization, Funding acquisition, Methodology, Project administration, Supervision, Writing – original draft, Writing – review and editing

### Author ORCIDs

Amanda H Lewis (iD) http://orcid.org/0000-0001-7316-9729
Jörg Grandl (iD) http://orcid.org/0000-0001-7179-7609

### Decision letter and Author response

Decision letter https://doi.org/10.7554/eLife.70988.sa1
Author response https://doi.org/10.7554/eLife.70988.sa2

## Additional files

### Supplementary files

• Transparent reporting form

### Data availability

Source data files have been provided with the numerical data for each figure.

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
