## [Decision Letter]

**Acceptance summary:**

The work provides important mechanistic detail on how Piezo channels, which are widespread and versatile mechanoreceptor molecules, functionally interact in the plane of the plasma membrane. It will be of interest to the field of mechanobiology and sensory mechanotransduction.

**Decision letter after peer review:**

Thank you for submitting your article "Piezo1 ion channels inherently function as independent mechanotransducers" for consideration by *eLife*. Your article has been reviewed by 3 peer reviewers, and the evaluation has been overseen by a Reviewing Editor and Richard Aldrich as the Senior Editor. The reviewers have opted to remain anonymous.

Essential revisions:

Lewis and Grandl propose that Piezo1 channels density does not have an effect on pressure sensitivity and that these channels do not cooperate in the nominal absence of membrane tension. To arrive at this conclusion the authors combined single channel measurements along with stochastic simulations of Piezo1 spatial distributions. An important element of this study is the use of two types of cells, one with intrinsically low level of Piezo1, and another with overexpressed channel. An interesting technical aspect is the use of a ramp pressure protocol, which overcomes the drawbacks of the standard step pressure method and help better estimate mechanosensitivity.

The topic of this manuscript is timely and relevant as the study of this family of ion channels is still very new and the details of their gating mechanism are unknown, and the central conclusion is important for understanding the physiological role of Piezo1 is various types of mechanically sensitive cells.

Although the manuscript is a tour de force, there are some concerns that should be addressed to validate the conclusions.

1) Patch imaging to validate conclusions. We note that the authors departed from their previous strategy of patch visualization and direct tension determination (Lewis and Grandl, *eLife* 2015). We do understand how tedious these experiments are, but the use of this approach in some of the trials would make this particularly study cleaner and more conclusive. The authors never mention tension as the parameter driving gating, and they work exclusively in the units of pressure. They postulated that their pipettes ranging between 3 and 6 MOhm in resistance provide 'standard' patches of given size and curvature. In fact, this range of resistances is large and roughly translates into a 1.4-fold difference in pipette diameters. If we look at Supplemental Figure 2D, we see that the largest pipettes (2-2.5 MOhm) produce 5 times the current recorded by 4-5 MOhm pipettes. These patches are unquestionably larger and no one can expect that the midpoint of channel activation by pressure is the same. The authors stated that only pipettes > 3MOhms were included into analysis, but this does not exclude actual size variation among patches. Figure 4F,G clearly shows that patches containing 0-20 channels have higher P50. It is very likely that these patches are smaller and therefore will require higher pressure for activation. We expect that imaging can help here. The authors do not need to cover the entire range of densities; only two sets of measurements with low (native) and high (overexpression) densities would be sufficient. We suggest that you choose reasonably large pipettes (2.5-3 MOhm) so you can reliably image cell-attached patches. Two sets of tension-Po curves, their tension midpoints and slope factors, recorded from HEK cells transfected at low and high expression levels will provide a clinching result.

2) Inactivated and silent channels. Because Piezo1 channels inactivate and require the removal of the stimulus to be reactivated, the presence of basal tension in the patch will render some Piezo1 inactivated. Given that the basal pressure differs between electrodes, different patches will always contain some number of 'silent' channels, which are physically present in the membrane, but do not manifest themselves functionally. This makes the estimation of a true number of channels in the patch virtually impossible using electrophysiology alone. The reviewers discussed this point at length. It was agreed that a few additional experiments (outlined below) would be ideal for mitigating all concerns on this issue. However, it was also agreed that the patch-imaging experiments requested above should be sufficient for supporting the paper's main conclusions, and therefore the experiments suggested below are suggested but not required. If the suggested experiments are not performed, we request that the authors mention / discuss the 'silent channel' point and indicate that their experiment provide a good estimation of the total number of channels, rather than the exact number.

Suggested experiments:

a) The authors claim that their novel stimulation protocol for single channel analysis allows them to measure the number of channels in each patch with high accuracy. The pressure ramps, used in this manuscript,¬ last about 3,000 ms at + 60mV. However, according to the same group (Wu et al., 2017, Cell Reports) Piezo1 inactivates within 90-100 ms at +60 mV. Hence, it is quite possible that in those patches the authors are underestimating the number of channels. It is not clear why the authors did not measure macroscopic currents using similar conditions as the ones used for single channel measurements. Particularly, macroscopic currents should be measured at positive potentials (to slow down inactivation), instead of at -80 mV, where Piezo1 tends to inactivate more rapidly than at positive potentials.

b) The authors should consider perfusing Yoda1 (Piezo1 specific agonist) at the end of the pressure ramp experiments to reveal that there are indeed no more channels in the patch membrane than those visible ones with pressure.

c) The representative traces on Figure 1B and 1D suggest that -60 mmHg does not saturate the response of Piezo1. The authors should add a control square pulse at the end of the experiment (equal to or larger than -60 mmHg) to ensure the currents are saturated.

3) Figure 2. The analysis performed on Figure 2 is confusing as is it is not clear how the mean open probabilities (black traces) as a function of pressure were calculated. Mean open and closed times need to be determined. Authors should elaborate when explaining these types of analyses and provide details on how good and/or significant these fits are.

4) Figure 4. It appears you have plotted the wrong data when reporting the slopes obtained by adjusting sigmoidal fits to their macroscopic data. The slopes don't have units, and instead they presented mmHg ranging from 0 to -25. Showing the sigmoidal fits and the slopes obtained from them might help driving the point.

5) Introduction. In the introduction, the authors propose that due to short-scale tension propagation in the membrane, the mechanosensors should be evenly distributed and be functionally independent. "Any substantial deviation from a uniform distribution of force-gated ion channels will result in domains that fall short to detect forces (where there are no ion channels) as well as domains that transduce forces disproportionately (where many ion channels are nearby)." In fact, the non-uniform distribution of Piezo channels has been shown by the Martinac group (Ridone et al., 2020). Additionally, Piezo1 recruitment to focal adhesions has been strongly suggested (Yao et al., https://www.biorxiv.org/content/10.1101/2020.03.09.972307). We suggest that you present a more balanced view on the potential sensitivity tuning of the 'receptive field' by channel clustering, and then ask whether it is the case.

6) We advise against insisting that the midpoint tension for Piezo1 is 1.4 mN/m. The fact that the patches bulge into the pipette at zero clamped pressure indicates that the actual pressure gradient is non-zero. It can be easily accounted for by the osmotic (oncotic) pressure difference between the cell interior and patch pipette buffer. The authors found that a positive pressure of 5 mm Hg flattens the patch and thus compensates for the existing pressure bias. The flat patch, like a drumskin, experiences the minimal (resting, seal-generated) tension. When the authors pre-condition the patch with +5 mm Hg and then stimulate with -20 mm Hg, the actual stimulus amplitude is -25 mm Hg. For this reason, the previously reported midpoint of ~5 mN/m for inside-out patches (less osmotic/hydrostatic gradients) might be more realistic. It is also more consistent with the work of other groups (Cox et al., 2016, 7:10366 | DOI: 10.1038/ncomms10366)

7) Why does the pressure go back to +5 mmHg the current goes beyond the baseline? This could indicate that channels were open before the ramp, suggesting the estimation might not be accurate.

---

## [Author Response]

Essential revisions:Although the manuscript is a tour de force, there are some concerns that should be addressed to validate the conclusions.1) Patch imaging to validate conclusions. We note that the authors departed from their previous strategy of patch visualization and direct tension determination (Lewis and Grandl, eLife 2015). We do understand how tedious these experiments are, but the use of this approach in some of the trials would make this particularly study cleaner and more conclusive. The authors never mention tension as the parameter driving gating, and they work exclusively in the units of pressure. They postulated that their pipettes ranging between 3 and 6 MOhm in resistance provide 'standard' patches of given size and curvature. In fact, this range of resistances is large and roughly translates into a 1.4-fold difference in pipette diameters. If we look at Supplemental Figure 2D, we see that the largest pipettes (2-2.5 MOhm) produce 5 times the current recorded by 4-5 MOhm pipettes. These patches are unquestionably larger and no one can expect that the midpoint of channel activation by pressure is the same. The authors stated that only pipettes > 3MOhms were included into analysis, but this does not exclude actual size variation among patches. Figure 4F,G clearly shows that patches containing 0-20 channels have higher P50. It is very likely that these patches are smaller and therefore will require higher pressure for activation. We expect that imaging can help here. The authors do not need to cover the entire range of densities; only two sets of measurements with low (native) and high (overexpression) densities would be sufficient. We suggest that you choose reasonably large pipettes (2.5-3 MOhm) so you can reliably image cell-attached patches. Two sets of tension-Po curves, their tension midpoints and slope factors, recorded from HEK cells transfected at low and high expression levels will provide a clinching result.

We thank the reviewers for this excellent suggestion! We performed this exact experiment and incorporated the new data, which all further strengthened the main conclusion.

Specifically, we combine our step electrophysiological protocol with DIC imaging in Neuro2a-p1ko cells overexpressing mouse Piezo1. In this dataset, we now independently measured radius for each pressure step, calculated membrane tension, and correlated this with Piezo1 open probability. We find that for channel densities ranging from 10-70 channels/mm2, there is no effect of channel density on tension midpoint (T_50_) or slope factor (k).

The experiments are illustrated in new Figure 5. As expected, this experiment also improved our estimates of surface density in these pipette resistance ranges (updated Figure 5—figure supplement 1). Associated text regarding these experiments is found in the updated Methods (lines 399-410), updated Results (lines 253-265), and updated Discussion (lines 311-313).

2) Inactivated and silent channels. Because Piezo1 channels inactivate and require the removal of the stimulus to be reactivated, the presence of basal tension in the patch will render some Piezo1 inactivated. Given that the basal pressure differs between electrodes, different patches will always contain some number of 'silent' channels, which are physically present in the membrane, but do not manifest themselves functionally. This makes the estimation of a true number of channels in the patch virtually impossible using electrophysiology alone. The reviewers discussed this point at length. It was agreed that a few additional experiments (outlined below) would be ideal for mitigating all concerns on this issue. However, it was also agreed that the patch-imaging experiments requested above should be sufficient for supporting the paper's main conclusions, and therefore the experiments suggested below are suggested but not required. If the suggested experiments are not performed, we request that the authors mention / discuss the 'silent channel' point and indicate that their experiment provide a good estimation of the total number of channels, rather than the exact number.Suggested experiments:a) The authors claim that their novel stimulation protocol for single channel analysis allows them to measure the number of channels in each patch with high accuracy. The pressure ramps, used in this manuscript,¬ last about 3,000 ms at + 60mV. However, according to the same group (Wu et al., 2017, Cell Reports) Piezo1 inactivates within 90-100 ms at +60 mV. Hence, it is quite possible that in those patches the authors are underestimating the number of channels. It is not clear why the authors did not measure macroscopic currents using similar conditions as the ones used for single channel measurements. Particularly, macroscopic currents should be measured at positive potentials (to slow down inactivation), instead of at -80 mV, where Piezo1 tends to inactivate more rapidly than at positive potentials.b) The authors should consider perfusing Yoda1 (Piezo1 specific agonist) at the end of the pressure ramp experiments to reveal that there are indeed no more channels in the patch membrane than those visible ones with pressure.c) The representative traces on Figure 1B and 1D suggest that -60 mmHg does not saturate the response of Piezo1. The authors should add a control square pulse at the end of the experiment (equal to or larger than -60 mmHg) to ensure the currents are saturated.

Again, we sincerely thank the reviewers for bringing up this important point, and agree that silent channels are likely present in patches. We have addressed this point with extensive additional experiments, quantifications, and text discussion:

First, in our original dataset assaying endogenous expression in Neuro2A cells (Figure 1), we had already included a control square pulse at the end of each experiment (as originally shown in Figure S2A), but recognize that we did not clearly explain the protocol, as well as the lack of inactivation for endogenous currents at +60 mV. To improve this, we have modified and/or added the following text:

Lines 77-78: “Consequently, a step protocol consistently overestimates current by 2-3 pA, and thus channel numbers accordingly.”

Lines 89-96: “We are confident that these slow changes in current are unrelated to Piezo gating, as currents clearly saturated after leak subtraction (Figure 1A-D, bottom). Additionally, we never observed channel openings during the ramp phase of the protocol in 15 patches from Neuro2A-Piezo1ko cells (data not shown). Ultimately, we decided to combine the ramp stimulus protocol with a subsequent brief (200 ms) saturating pressure step to -60 mmHg. Importantly, the step pulse allowed us to confirm post-hoc that inactivation of Piezo1 at this voltage is minimal (step pulse: mean current at 200 ms = 96.5±7.4% of peak; n = 281 patches; Figure 1 —figure supplement 2 A,D).”

Second, we agree with the reviewers that the overexpression experiments are better done at +60 mV. Consequently, we performed a completely new experiment at this positive voltage.

We find in this new, extensive dataset (144 patches) that there remains no correlation between P50 and channel density. We also find that on average, our estimates of channel density have roughly doubled (from a mean of 40 channels/mm^2^ when measured at -80 mV to 90 channels/mm2 when measured at +60 mV). This is consistent with the reviewers’ intuition that silent channels are present. Importantly, this new experiment also confirms that we have covered an even higher range of Piezo1 channel densities than we previously estimated, giving us further confidence that if there were a cooperative effect of channel density on gating properties, we would have observed it.

The experiments are described in new Figure 4, and updated Figure 4 —figure supplement 1 and Figure 4—figure supplement 4. Updated text associated with this experiment is found in the updated Result (lines 194-252). We have also better highlighted the contribution of silent channels in several places in the text:

Lines 228-230: “Moreover, this calculation likely represents a slight underestimate of the true channel density, as residual inactivation at positive potentials may lead to a population of “silent” channels that do not manifest themselves functionally.”

Lines 289-292: “Assessing Piezo1 expression levels is also subject to error due to the presence of inactivation, which prevents reaching a P_o_ = 1. Specifically, channels inactivating during the rising phase of the current, as well as the presence of “silent” channels that fail to respond to the stimulus entirely, contribute to this error.”

3) Figure 2. The analysis performed on Figure 2 is confusing as is it is not clear how the mean open probabilities (black traces) as a function of pressure were calculated. Mean open and closed times need to be determined. Authors should elaborate when explaining these types of analyses and provide details on how good and/or significant these fits are.

We have modified/added associated text:

Lines 443-447: “To generate idealized currents (Figure 2), the number of channels (n) in each patch, as well as the time of openings, were manually identified. The open probability at a given time was taken as the fraction of total channels in the patch that were open. Owing to the lack of inactivation of endogenous currents at +60 mV, as well as the continually increasing pressure, channel closings were rare.”

Additionally, we added overlays of fits to the mean data (red dashed lines) in updated Figure 2.

4) Figure 4. It appears you have plotted the wrong data when reporting the slopes obtained by adjusting sigmoidal fits to their macroscopic data. The slopes don't have units, and instead they presented mmHg ranging from 0 to -25. Showing the sigmoidal fits and the slopes obtained from them might help driving the point

The sigmoidal fit function we use has slope on the bottom of the exponential (P_50_ – P)/k, such that its units are in mmHg. We now detail this in the updated Methods (lines 417-420):

“Current-pressure curves were fit with a Boltzmann function

I= Imin+ Imax1+e(P50−Pk)

where I_max_ is the maximal current, P is pressure, P_50_ is pressure of half-maximal activation, and k is the slope factor.”

5) Introduction. In the introduction, the authors propose that due to short-scale tension propagation in the membrane, the mechanosensors should be evenly distributed and be functionally independent. "Any substantial deviation from a uniform distribution of force-gated ion channels will result in domains that fall short to detect forces (where there are no ion channels) as well as domains that transduce forces disproportionately (where many ion channels are nearby)." In fact, the non-uniform distribution of Piezo channels has been shown by the Martinac group (Ridone et al., 2020). Additionally, Piezo1 recruitment to focal adhesions has been strongly suggested (Yao et al., https://www.biorxiv.org/content/10.1101/2020.03.09.972307). We suggest that you present a more balanced view on the potential sensitivity tuning of the 'receptive field' by channel clustering, and then ask whether it is the case.

To achieve this we included the suggested references and updated the Introduction as follows:

Lines 12-14: “The degree to which forces are transduced homogeneously across the cell membrane likely depends not only on the dynamics of membrane tension propagation, but also on (i) the spatial distribution of force-gated ion channels, and (ii) on their functional independence.”

Line 20: “Indeed, there is recent experimental evidence suggesting that Piezo1 channels may be non-uniformly distributed… Additional studies have proposed the existence of clusters with varying numbers of channels (Jiang et al., 2021), and that Piezo1 may be concentrated at focal adhesions (Mingxi Yao, 2020).”

6) We advise against insisting that the midpoint tension for Piezo1 is 1.4 mN/m. The fact that the patches bulge into the pipette at zero clamped pressure indicates that the actual pressure gradient is non-zero. It can be easily accounted for by the osmotic (oncotic) pressure difference between the cell interior and patch pipette buffer. The authors found that a positive pressure of 5 mm Hg flattens the patch and thus compensates for the existing pressure bias. The flat patch, like a drumskin, experiences the minimal (resting, seal-generated) tension. When the authors pre-condition the patch with +5 mm Hg and then stimulate with -20 mm Hg, the actual stimulus amplitude is -25 mm Hg. For this reason, the previously reported midpoint of ~5 mN/m for inside-out patches (less osmotic/hydrostatic gradients) might be more realistic. It is also more consistent with the work of other groups (Cox et al., 2016, 7:10366 | DOI: 10.1038/ncomms10366)

We removed the exact value for tension from the text (lines 5-7):

“For example, Piezo1 is a mechanically gated cation channel that directly senses membrane tension (T) with high sensitivity (Coste et al., 2010; Cox et al., 2016; Lewis and Grandl, 2015; Syeda et al., 2016).”

7) Why does the pressure go back to +5 mmHg the current goes beyond the baseline? This could indicate that channels were open before the ramp, suggesting the estimation might not be accurate.

We believe this change in baseline current is related to changes in leak, owing to recording at positive potentials, because in these solutions we frequently notice that the leak continuously improves at +60 mV, such that after the pressure step, the baseline current is less than before. This is true even for patches with zero channels (Figure 1A) and unrelated to Piezo1 gating as the effect can be observed to the same extent in N2A-P1ko cells.